# Protocol for a human in vivo model of acute cigarette smoke inhalation challenge in smokers with COPD: monitoring the nasal and systemic immune response using a network biology approach

Clare L Ross,[1] Neil Galloway-Phillipps,[2] Paul C Armstrong,[3] Jane A Mitchell,[2] Timothy D Warner,[3] Christopher Brearley,[4] Mari Ito,[5,6] Tanushree Tunstall,[1] Sarah Elkin,[1] Onn Min Kon,[1] Trevor T Hansel,[1] Mark J Paul-Clark[2]

TTH and MJP-C contributed equally.

For numbered affiliations see end of article.

**Correspondence to**
Dr Trevor T Hansel;
t.hansel@imperial.ac.uk

## ABSTRACT

**Introduction:** Cigarette smoke contributes to a diverse range of diseases including chronic obstructive pulmonary disease (COPD), cardiovascular disorders and many cancers. There currently is a need for human challenge models, to assess the acute effects of a controlled cigarette smoke stimulus, followed by serial sampling of blood and respiratory tissue for advanced molecular profiling. We employ precision sampling of nasal mucosal lining fluid by absorption to permit soluble mediators measurement in eluates. Serial nasal curettage was used for transcriptomic analysis of mucosal tissue.

**Methods and analysis:** Three groups of strictly defined patients will be studied: 12 smokers with COPD (GOLD Stage 2) with emphysema, 12 matched smokers with normal lung function and no evidence of emphysema, and 12 matched never smokers with normal spirometry. Patients in the smoking groups are current smokers, and will be given full support to stop smoking immediately after this study. In giving a controlled cigarette smoke stimulus, all patients will have abstained from smoking for 12 h, and will smoke two cigarettes with expiration through the nose in a ventilated chamber. Before and after inhalation of cigarette smoke, a series of samples will be taken from the blood, nasal mucosal lining fluid and nasal tissue by curettage. Analysis of plasma nicotine and metabolites in relation to levels of soluble inflammatory mediators in nasal lining fluid and blood, as well as assessing nasal transcriptomics, ex vivo blood platelet aggregation and leucocyte responses to toll-like receptor agonists will be undertaken.

**Implications:** Development of acute cigarette smoke challenge models has promise for the study of molecular effects of smoking in a range of pathological processes.

**Ethics and dissemination:** This study was approved by the West London National Research Ethics Committee (12/LO/1101). The study findings will be

**Strengths and limitations of the study**

- This model involves limited numbers of highly selected individuals, and as such selection bias will inevitably occur. However, in conjunction with serial sampling in controlled conditions, this enables discrimination of relatively small changes in levels of cigarette smoke constituents and the subsequent inflammatory response.
- It is possible to study acute effects of cigarette smoke in a variety of smoking-related diseases. In this way it may be possible to gain molecular insight into the pathology of chronic obstructive pulmonary disease, cardiovascular disease and neoplasia.
- The specific categorisation of patients with notable emphysema, who are not currently taking any anti-inflammatory or disease modifying interventions, will help reveal target pathway nuances that drive oxidant-dependent chronic disease.

presented at conferences and will be reported in peer-reviewed journals.

## BACKGROUND

There are profound causative and contributory effects of chronic cigarette smoking in a range of disease processes, as well as considerable mortality.[1 2] In order to understand the changes in the lungs due to chronic smoking, it is relevant to assess the immunological responses to acute smoke exposure. Repetitive acute effects of cigarette smoke in susceptible individuals may lead to cumulative irreversible damage.[3] Despite cigarette smoking being the

most important risk factor for the development of chronic obstructive pulmonary disease (COPD) and a major contributory influence in the development of cardiovascular disease, the use of human cigarette challenge models has been underutilised in the investigation of immune and local influences on the disease.[4] This is somewhat surprising, as we still know little about the mechanisms involved in smoke-induced inflammation in man.

While several downstream effects of cigarette smoking are common to all smokers, such as antioxidant gene activation and aryl hydrocarbon signalling, it is estimated that only 13–50% of smokers actually develop COPD,[5] [6] and within this group, there is great clinical heterogeneity.[2] The immunopathology of COPD is complex and variable, involving the large airways (bronchitis), small airways (bronchiolitis), lung interstitium (emphysema and interstitial lung disease), pulmonary vasculature (pulmonary artery hypertension) and systemic and cardiovascular complications.[7] There is also the added complexity attributable to the innate immune response to oxidants and microbes.[8] [9] Our group and others have previously demonstrated that cigarette smoke (extract) can activate human immune and respiratory epithelial cells in vitro leading to the release of the proinflammatory chemokine CXCL8.[10–15] In addition, following smoking, blood of smokers is 'primed' to activation ex vivo by pathogen-associated molecular patterns (PAMPs) including lipopolysaccharide (LPS).[16] Most recently we have performed a pilot transcriptomic study using human monocytes stimulated in vitro and shown that smoke activates and inhibits discrete groups of genes involved in oxidant stress and inflammation.[17] In addition there have been individual genomic, transcriptomic and metabolomic studies in cell-based[18–24] and animal models[25] of smoking which have further defined the role of cigarette smoke as an inflammatory insult. However, there is now a need for a multisystems-based approach in man in vivo to truly advance our understanding of how cigarette smoking induces inflammation.[26]

A protocol for a study of the effects of smoking in patients with COPD has recently been reported by Lo Tam Loi et al[27] from Utrecht, Netherlands. This group proposes the assessment of acute effects of smoking at 5 min after smoking three cigarettes, at 2 h, 24 h and after a 6 week interval. They also propose assessment of cross-sectional inflammatory responses in different patient groups. In this study sampling from patients consists of blood, sputum and exhaled breath condensate (EBC); and they employ endobronchial sampling for biopsy, epithelial lining fluid and epithelial brushings.

Our group is currently conducting an in vivo model of acute cigarette smoke inhalation challenge in smokers with COPD and appropriate controls. Our study differs in selection of patients, and having more defined conditions for cigarette smoke exposure. In addition, we have an intensive sampling schedule over the 5 h following a controlled cigarette smoke stimulus, with a focus on blood and nasal non-invasive sampling, during which we assess levels of nicotine and metabolites in relation to proinflammatory effects. Such a study has thus far been difficult since access to human airway tissue and secretion samples in a minimally invasive serial manner has not previously been possible.

There has been recent progress in finding novel biomarkers for COPD,[28] [29] and a focus on recognising new phenotypes of COPD.[30] GlaxoSmithKline has completed a 3-year longitudinal study in 2180 patients with COPD entitled ECLIPSE (Evaluation of COPD Longitudinally to Identify Predictive Surrogate Endpoints).[31] In terms of the natural history of COPD it was found that sputum neutrophil counts[32] and EBC pH was not useful.[33] EBC has limitations of dilution and salivary contamination.[34] In contrast, induced sputum contains many dead and dying cells, making quantitation of levels of inflammatory mediators problematic.

The nasal epithelium is the first point in the respiratory system where cigarette smoke has contact with the respiratory mucosa. As part of 'the one airway concept' that is well established in asthma, there is also considerable evidence for nasal involvement in COPD.[35] [36] Patients with COPD have chronic nasal symptoms and impaired quality of life,[37–39] with upper and lower airway inflammation,[40] and exacerbation of COPD is associated with increased pan-airway inflammation.[41] In addition, young 'healthy smokers' have functional and inflammatory changes in the nose and lower airways.[42]

For this reason our study is based on taking respiratory samples from the nose by nasal absorption and curettage. It has long been recognised that there is 'one airway', with a strong functional and immunological relationship between the nose and the bronchi.[43–45] Patients with respiratory disease commonly have inflammation of the airways and nasal passages, with a similar inflammatory infiltrate in the lower and upper airways. It is now possible to obtain repeated samples of nasal exudates before and after nasal challenge in a relatively non-invasive manner by techniques employing strips of nasal synthetic absorptive matrix (SAM) inserted into the nostril in the technique of nasosorption. Our experience with SAM for nasosorption has been published with regard to cytokines and chemokines in children with allergic rhinitis,[46] infants with a family history of atopy,[47] and in atopic adults after nasal allergen challenge.[48] In addition, nasal epithelial curettage employing the Rhinoprobe device is useful to obtain a pinhead of mucosal tissue, in a technique that does not require local anaesthesia.[49] This approach has the advantages of being safe and comfortable for the patient, yet providing high-quality samples for transcriptomic analysis.

It is well documented that cigarette smoke is a complex stimulus with a variety of acute and chronic effects reported in the literature. We set out to design a study to map the acute inflammatory response to smoke in the human respiratory system and circulating cells, with a view to providing a comprehensive molecular signature of smoking-related events in COPD. We believe

that our model of cigarette smoke exposure complements the clinical research model of Lo Tam Loi *et al*.[27] These approaches should lead to advances within the field of assessment of smoking-related immunopathology; therefore we have taken this opportunity to share our rationale and protocol.

## Aims of the study

Our study uses non-invasive techniques for sampling, to which 'omic' technology will be utilised for the comprehensive characterisation of this complex multifaceted disease, with the aim of identifying disease-dependent whole system responses to acute cigarette smoke challenge.[50] The transcriptomic analysis in this study has been designed to examine changes in gene expression under the specific physiological condition of acute cigarette smoking. This may in future allow for early intervention in populations exhibiting similar gene expression profiles to those observed in established COPD. This approach also provides a diagnostic profile of patients so that treatment can be targeted and personalised. Theoretically, it may be possible to subclassify COPD populations, who phenotypically appear similar. In addition, there is a need to analyse products of these genes, given that transcriptomic analysis does not solely account for the diversity in protein production and cellular metabolites. A range of cytokines will be measured before and after an acute cigarette challenge to identify biomarkers and cell-signalling pathways associated with COPD. Metabolic profiling will be used to detect the physiological changes induced by a cigarette challenge. Metabolic signatures may provide prognostic, diagnostic and surrogate markers for COPD, and identify simple non-invasive markers of drug responses for future therapies.[51] These investigations will be carried out on nasal lining fluid, nasal curettage, and blood, to analyse local and systemic changes over a 5 h period following a two cigarette challenge. Finally, it has been suggested that patients with stable COPD have increased platelet reactivity including circulating platelet-monocyte aggregates,[52] therefore, potential cardiovascular effects will also be assessed by measuring platelet aggregation in these patients.

## Primary objective

To develop a novel cigarette challenge in vivo model incorporating full network biology analysis of transcriptomic, metabolomic and cytokine/chemokine changes in the nose and blood of smokers with Global Initiative for Chronic Obstructive Lung Disease (GOLD, http://www.goldcopd.org) Stage 2 COPD, healthy smokers with normal lung function and non-smokers, post-cigarette or post-sham/dummy cigarette challenge.

## Secondary objectives

1. To stimulate blood of smokers with GOLD Stage 2 COPD[16] healthy smokers and non-smokers ex vivo with interleukin-1β (IL-1β) and PAMPs.

2. To identify molecular biomarkers for patients with COPD: to assist in defining novel therapeutic targets, to better stratify phenotypes and to facilitate monitoring of patients.

3. To develop a cigarette smoking challenge model in patients with COPD with an aim to utilise this in therapeutic trials of novel therapeutic agents.

4. To carry out platelet aggregometry following stimulation with specific agonists, with the aim of understanding the associated pathophysiology of thrombosis and the pharmacology of respective therapies.

## METHODS

### Study populations

This is a parallel group study in three groups of 12 age, sex, ethnicity, smoking history and body mass index matched patients (table 1):

▶ Group 1: Smokers with moderate COPD (GOLD Stage 2);
▶ Group 2: Healthy smokers with normal lung function (no evidence of COPD);
▶ Group 3: Healthy individuals who have never smoked.

Smokers will be current smokers, smoking at least five cigarettes a day, with a minimum pack year history of 10 years. Current cannabis smokers or smokers with a history of moderate or heavy cannabis use will be excluded from the study. Non-smokers must not have smoked a single cigarette in the 12 months prior to the study, and must have smoked less than 100 cigarettes in their lifetime.

### Cigarette smoke challenge procedure

One to three screening visits may be required to complete spirometry (and full lung function with gas transfer for carbon monoxide (TLCO) in smokers), 5 slice high-resolution CT (in smokers only) as well as laboratory safety tests, urinalysis, ECG and physical examination.

On the cigarette challenge day, all patients will be required to attend our unit at 9:00. They must have fasted and refrained from smoking from 21:00 on evening prior to the scheduled challenge. Following baseline investigations, smokers in groups 1 and 2 will smoke two cigarettes back to back, in a controlled environment, exhaling the smoke via their nostrils, while non-smokers carry out normal tidal breathing over a 10 min period (figure 1).

### Schedule of sampling

All patients will have nasal epithelial curettage and nasosorption procedures with serial blood samples (figures 2 and 3). Serial nicotine and cotinine levels will also be taken to plot the relative smoke exposure of each individual, as well as providing an objective measure of their baseline smoking habit and clearance of nicotine from their system. Full blood counts and clotting studies will also be performed during the study to ensure patients

**Table 1** Summary of inclusion criteria and assessments in the acute cigarette smoke challenge

| Inclusion criteria | Sample and assessment parameters | References |
|---|---|---|
| General criteria for 3 groups of patients, each of 12 individuals<br>▶ 45–75 years old<br>▶ Good general health, with no chronic illnesses<br>▶ No prescribed anti-inflammatory medications (including statins)<br>▶ Females of childbearing potential have a negative pregnancy test | *Plasma*Cigarette smoke<br>Correlates of exposure<br>▶ Nicotine<br>▶ Cotinine<br>▶ 3-Hydroxycotinine<br>Plasma mediators<br>▶ Prostanoids<br>▶ Metabolites<br>▶ Cytokines and chemokines<br>Whole blood ex vivo<br>▶ Transcriptomics<br>▶ TLR-agonist stimulation of leukocytes<br>▶ Platelet aggregation | Nicotine[54]<br>Serum cytokines[29 55–59]<br>Serum metabolites[60–62]<br>Blood ex vivo stimulation[16 17] |
| Smokers<br>▶ Current: ≥5 cigarettes/day<br>▶ History: ≥10 pack years<br>Non-Smokers<br>▶ Have never smoked<br>Group 1: Smokers with GOLD Stage 2 COPD<br>▶ Post-bronchodilator $FEV_1$ 50–79%; Forced expiratory ratio <70%<br>▶ TLCO of <80% of normal<br>▶ Emphysema on 5 slice HRCT of the chest<br>Group 2: Healthy smokers<br>▶ Post-bronchodilator $FEV_1$ ≥80%; Forced expiratory ratio ≥70%<br>▶ TLCO ≥80% of normal<br>▶ Normal 5 slice HRCT of the chest | Nasosorption (SAM)<br>▶ Prostanoids: LTB4, LTC4, PGD2<br>▶ Metabolites<br>▶ Cytokines and chemokines<br><br>Nasal curette<br>▶ Immunohistology<br>▶ Transcriptomics<br>▶ Flow cytometry<br>Epithelial culture<br>EBC | EBC[33 63] |
| Group 3: Healthy non-smokers<br>▶ Post-bronchodilator $FEV_1$ ≥80%; Forced expiratory ratio ≥70%<br>▶ TLCO and HRCT not done | Sputum | Sputum[32 64 65] |

COPD, chronic obstructive pulmonary disease; EBC, exhaled breath condensate; $FEV_1$, forced expiratory volume in one second; HRCT, high-resolution CT; SAM, synthetic absorptive matrix; TLR, toll-like receptor.

**Figure 1** Cigarette smoke challenge model (HEPA, high-efficiency particulate air).

All subjects are asked to refrain from smoking for 12h before and 5h after cigarette challenge.
All subjects are fasted and permitted to drink water for 12h before and 5h after cigarette challenge.

**Groups 1 & 2: Smokers**
On the morning of the challenge, following baseline investigations, each subject is asked to smoke 2 Marlboro Red cigarettes, one after the other, over a 10 min period in a ventilated chamber. They inhale normally, but exhale only through the nose.

**Groups 3: Non-Smokers**
On the morning of the challenge, following baseline investigations, each subject is asked to breath normally for 10 min in a smoke-free environment.

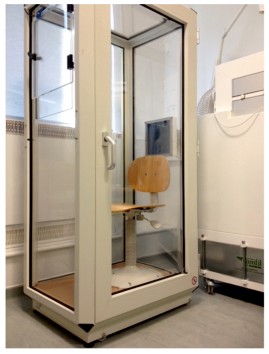

Ventilated chamber with carbon filter attached to ventilation unit in addition to HEPA filter

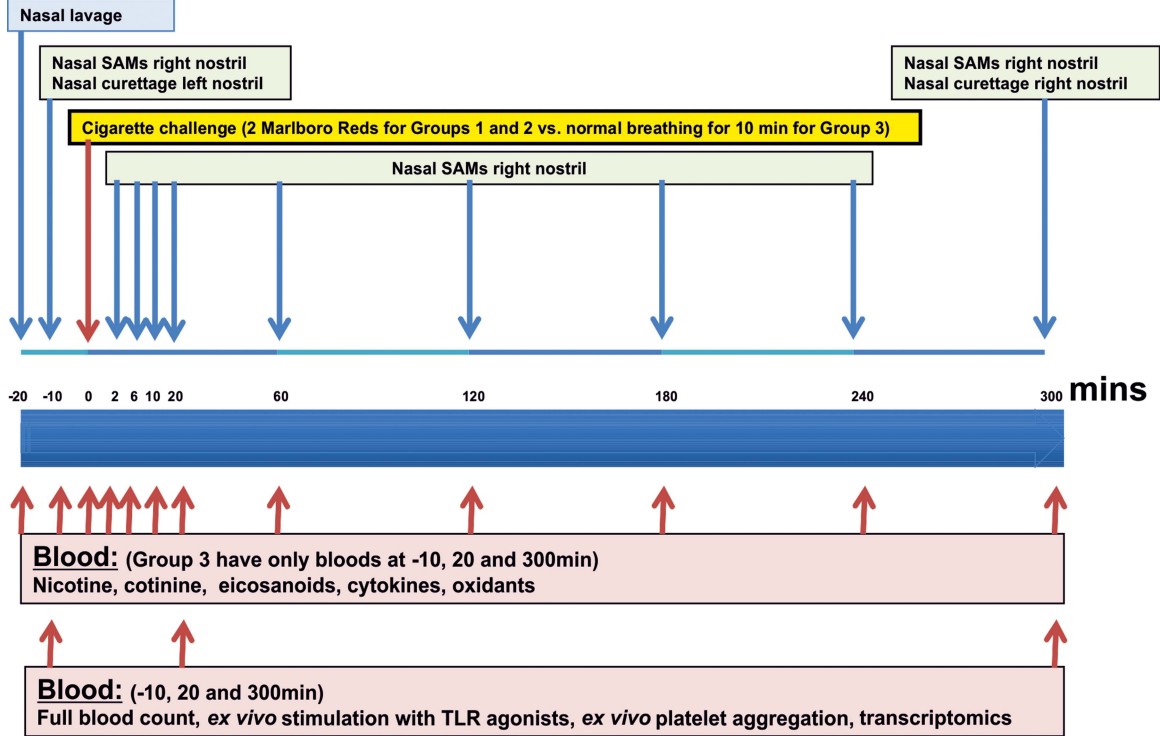

**Figure 2** Schedule of assessments (SAM, synthetic absorptive matrix, TLR, toll-like receptor).

have not developed any biochemically relevant illnesses or clotting abnormalities.

## Analytical methods

A representative range of non-invasive sampling methods with associated analytical parameters is shown in table 1.

### Nicotine/cotinine

Nicotine, cotinine and 3-hydroxy-cotinine will be measured in serum over the whole time course. Analysis by capillary gas chromatography will be carried out by Advanced Bioanalytical Service Laboratories (Welwyn Garden City, UK).

### 8-Isoprostane enzyme immunoassay

Measurement of 8-isoprostane will be carried out in both serum and nasoabsorption fluid at all time points using an 8-isoprostane enzyme immunoassay (EIA) kit (Cayman Chemicals, Ann Arbor, Michigan, USA).

### Metabolomics

In view of the multiple factors that can influence metabolism, all patients are required to fast for 12 h before the challenge begins; during this time they may only consume water. Metabolomic profiling, to be conducted by Metabolon (North Carolina, USA), measures an extensive range of metabolites (<1000 Da) in plasma and nasal

**Figure 3** Nasal sampling methods (SAM, synthetic absorptive matrix).

**Nasosorption using SAM**

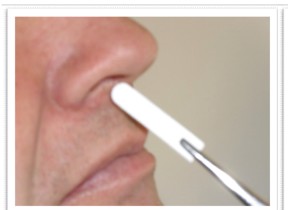 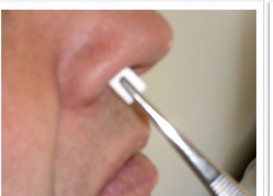 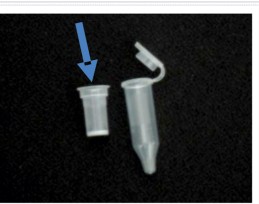

**Nasal curettage using a Rhino-probe**

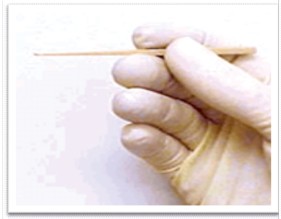 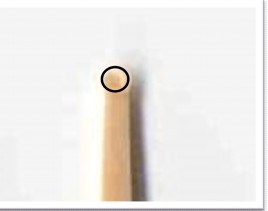

lining fluid after cigarette smoke. Metabolomic analysis will include amino acids, carbohydrates, lipids, nucleic acids and cofactors, molecules of redox homeostasis (eg, glutathione), organic acids, and small peptides. Importantly, many of the catabolites or biosynthetic intermediates of these metabolites are also detected, assisting in elucidating underlying mechanistic insight. Blood and nasoabsorption samples will be taken at time points listed in figure 2.

### Transcriptomics

Two nasal epithelial curettage samples will be taken using a Rhinprobe prechallenge and at 300 min postchallenge. RNA will be extracted using TRIzol (Invitrogen, Paisley, UK). Blood will be taken at prechallenge, 20 and 300 min postchallenge with RNA extracted using the PAXgene Blood RNA extraction kit (PreAnalytiX GmbH, Hombrechtikon, Switzerland). RNA will be run on an Illumina HT12 V.4 (RefSeq Build 38 Rel 22) chip (Illumina, San Diego, USA) with analysis carried out using a dedicated array analysis programme GeneSpring GX 11.3 of all genes. Genes which have changed significantly will be identified using unpaired t tests ($p < 0.05$) with Benjamini-Hochberg false discovery rate correction. Genes will be reported as 1.2-fold, 1.5-fold and twofold increases. Quality and assessment of global transcriptome changes will be assessed using principle component analysis in GeneSpring. Genes that have changed significantly will also be further explored using a dedicated pathway analysis tool (Ingenuity Systems Pathway Analysis software, IPA) techniques.

### Ex vivo peripheral whole blood stimulation

Stimulation of whole blood taken prechallenge and at 20 and 300 min postchallenge will be performed using a range of toll-like receptor (TLR) agonists to obtain a 24 h dose response curve of the following ligands: LPS (TLR4), FSL-1 (TLR6/2), Pam3CSK4 (TLR1/2), Poly(I:C) (TLR3) and IL-1β (Invivogen, San Diego, USA). Serum will then be removed and subsequently measured for CXCL8 and IL-1β by ELISA (R&D Systems, Abingdon, UK).

### Platelet aggregation

For platelet aggregation studies, blood will be collected prechallenge and at 20 and 300 min postchallenge. Platelet rich plasma will be aliquoted into individual wells of half-area 96-well plates coated with gelatin and one of seven concentrations of arachidonic acid, ADP, collagen, epinephrine, ristocetin, TRAP-6 amide or U46619. Platelet aggregation will be determined by changes in light absorbance, and release of thromboxane (TX)A2 by ELISA.

### Homogeneous time resolved fluorescence assay

Serum and nasoabsorption fluid will be screened, across all time points, for prostaglandin E2 and leukotriene B4 levels using the homogeneous time resolved fluorescence (HTRF) assay kits from Cisbio Assays (Bedford, MA).

### Chemokine/cytokine immunoassay

Using a Meso Scale Discovery (MSD) immunoassay system (MSD, Maryland, USA), a variety of chemokines, cytokines and vascular markers will be measured in blood and nasoabsorption samples at the time points list above.

### Statistical analyses

This is an exploratory clinical study, and we are aware that there may not be detectable differences between groups based on measurement of particular parameters and the size of effects.

## DISCUSSION

Clinical challenge models have been fundamental to clinical research in asthma: employing inhalation of agents such as methacholine, histamine, AMP, allergens and occupational agents such as isocyanate and ozone. In contrast there has been little clinical research on the effects of cigarette smoke in vivo involving patient studies, despite this being the known causative agent in COPD. Development of clinical challenge models that involve cigarette smoke thus have relevance to studying respiratory, cardiovascular and neoplastic effects of cigarette smoke.

The major consideration in developing a cigarette smoke challenge model is the ethical aspect of not encouraging a smoker to continue smoking, and ensuring that maximal support is given to the individual to stop smoking (table 2). Furthermore, we are studying patients with mild–moderate disease, in whom there may be the possibility of smoking cessation before permanent disability sets in. Other studies have tended to evaluate later stage disease, by which time smoking cessation has less beneficial effects in terms of lung function.[53]

A secondary consideration is ensuring the well-being of the scientific and clinical staff involved in the study, and minimising exposure to cigarette smoke. In our study design, all smokers will be established in the habit and will be actively encouraged to enter into a smoking cessation programme immediately following their cigarette challenge. Our unit has adapted a body plethysmography box, with the addition of a carbon filter and high-efficiency particulate air filter, in order to ensure staff are not exposed to the harmful effects of smoke. If patients were to receive their cigarette challenge outside of the hospital, there may be confounding effects of additional pollutants, temperature and exercise.

An important feature of the study is that smoking two cigarettes is physiologically relevant as a challenge, and that we were able to document levels of nicotine and metabolites over a 5 h period. We adopted this aspect of the study having considered the design of pharmacokinetic studies with nicotine delivery devices. We intend to standardise the technique of smoking by giving two Marlboro Red cigarettes, noting the total number of inhalations and encouraging exhalation through the nose. Controlled smoke exposure enables accurate

**Table 2** Features and ethical issues with the acute cigarette smoke challenge

| Features of the model | Ethical issues |
|---|---|
| Advantages<br>▸ Human model<br>▸ Acute-on-chronic inflammation<br>▸ Serial non-invasive sampling<br>▸ Combined direct measurement of biomarkers and ex vivo stimulation<br>▸ Limited numbers of strictly defined patients<br>▸ Compare with in vivo animal models<br>Disadvantages<br>▸ Difficulty recruiting a small number of highly defined patients<br>▸ Need to validate upper versus lower airway inflammation, including tissue biopsies<br>▸ Signal parameters must reliably change after acute cigarette smoke exposure<br>▸ Lung function and CT changes may occur after acute smoke exposure | All patients must be advised to stop smoking, and offered full clinical, psychological and pharmacological support to carry this out<br>There must be no encouragement for the patient to begin or continue smoking<br>Some frail patients with COPD will have difficulty fasting and refraining from cigarettes for the morning<br>Clinical disease detected through the investigations must be fully treated, regardless of participation in the study<br>Some frail patients with COPD will have difficulty fasting and refraining from cigarettes for the morning<br>The patient should not be taking any anti-inflammatory or confounding therapy: therapy must not be withheld |

COPD, chronic obstructive pulmonary disease.

assessment of patients' smoking exposure by measuring concomitant nicotine and cotinine levels. The majority of studies do not mandate any particular smoking restrictions prior to sampling. In this study it may be possible to formally compare nicotine exposure with levels of induced biomarkers.

A key feature of our acute cigarette smoke challenge model involves fasting and refraining from smoking for 12 h before and 5 h after having a controlled cigarette smoke exposure. We also ensure that patients are not taking any medication that may interfere with responses. This abstinence is necessary due to the extreme sensitivity of measurements such as metabolomics. Metabolomics involves assessment of levels of small molecules and will include molecules such as dietary constituents and drugs. We take serial blood and nasal samples, in a manner

similar to a phase I pharmacokinetic study of exposure to a single dose of drug.

The proposed study involves precision nasal sampling. This is non-invasive, and has potential for point-of-care, bedside and clinic monitoring. In contrast, bronchoscopy is a research procedure only undertaken with great care in patients with COPD. Hence, non-invasive sampling offers great potential for future use of the model in further observational and drug studies. In contrast to many studies looking at gene expression in cross-sectional populations of smokers, our model has the benefit of acquiring samples longitudinally before and after a challenge with a known trigger of the disease. This increases the power to detect effects of the cigarette smoke challenge. We will thoroughly evaluate the acute response to cigarettes with 10 blood and nasal sampling time points within a 5 h challenge period.

**Figure 4** Human integrated iterative inflammometry (CVS, cardiovascular system; TLR, toll-like receptor; PAMPs, pathogen-associated molecular patterns).

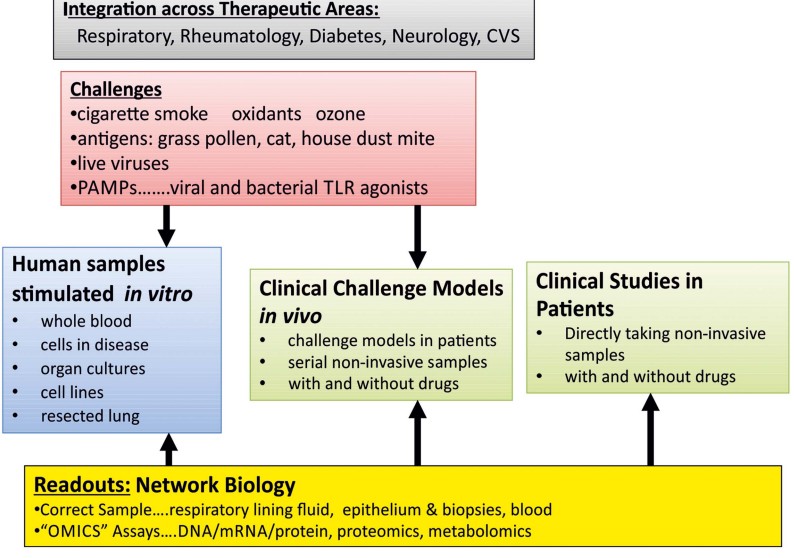

Given the heterogeneity of the disease and the fact that patients with significant comorbidities (such as cardiovascular disease) are excluded, we are likely to be evaluating a subpopulation of patients with mild COPD. This is inevitable in any COPD study, but worth noting, as it may be that this group behaves differently to those with comorbidities.

The setting up of such studies, using a fully integrated approach that incorporates network biology, where sampling occurs after the administration of causative factors will help us better understand the disease and design more robust clinical trials. This type of approach is illustrated in figure 4 has a far ranging applications to a number of chronic inflammatory diseases.

**Author affiliations**
[1]Imperial Clinical Respiratory Research Unit (ICRRU) and Biomedical Research Centre (BMRC), Centre for Respiratory Infection (CRI), St Mary's Hospital, Imperial College, London, UK
[2]National Heart and Lung Institute, Imperial College, London, UK
[3]William Harvey Research Institute, Barts and The London, Queen Mary's School of Medicine and Dentistry, London, UK
[4]Sunovion Pharmaceuticals Europe Ltd. (Sunovion Europe), London, UK
[5]Dainippon Sumitomo Pharma Co Ltd, Osaka, Japan
[6]Department of Molecular Regulation for intractable Diseases, Institute of Medical Science, Tokyo Medical University, Tokyo, Japan

**Contributors** CLR contributed to designing the clinical aspect of the study and writing the manuscript. NG-P contributed to the ethics submission and design of laboratory-based methods for the protocol. PCA contributed to the design of the platelet protocols, ethics submission and writing the manuscript. JAM contributed to the writing of the manuscript and design of the overall study. TDW contributed to the design of the platelet protocols and ethics submission. CB contributed to writing of the ethics, over all protocol design and its compliance with GCP. MI, SE, OMK and TT contributed to the design of the clinical protocol, ethics submission and manuscript composition. TTH was the main author, contributed to the design of the overall protocol and ethics submission. MJP-C contributed to the design of the overall protocol, design of the assays, ethics submission and writing the manuscript.

**Funding** This study was funded by the Wellcome Trust and Dainippon Sumitomo Pharma Co Ltd, Osaka, Japan. Supported by: Dainippon Sumitomo Pharma Co Ltd, Osaka, Japan National Institute of Healthcare Research (Grant No: R3101002), NIHR Imperial Biomedical Research Centre (NIHR BMRC), Imperial Academic Health Science Centre (AHSC), Imperial Centre for Respiratory Infection (CRI, Grant No: 083567/Z/07/Z), Wellcome Trust (Grant No: 083429/Z/07/Z).

**Competing interests** None.

**Ethics approval** West London National Research Ethics Committee.

**Provenance and peer review** Not commissioned; externally peer reviewed.

**Data sharing statement** We are happy to share all resorces and protocols defined in this paper with others in the field or the larger scientific community.

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
