## [Reviewer comments · BMJ Open]

ARTICLE DETAILS

TITLE (PROVISIONAL)	Protocol for a human in vivo model of acute cigarette smoke inhalation challenge in smokers with COPD: monitoring the nasal and systemic immune response using a network biology approach
AUTHORS	Ross, Clare; Galloway-Phillipps, Neil; Armstrong, Paul; Mitchell, Jane; Warner, Timothy; Brearley, Christopher; Ito, Mari; Tunstall, Tanushree; Elkin, Sarah; Kon, Onn Min; Hansel, Trevor; Paul-Clark, Mark

VERSION 1 - REVIEW

REVIEWER	Renat Shaykhiev Weill Cornell Medical College
REVIEW RETURNED	23-Jul-2014

GENERAL COMMENTS	The manuscript by Ross and co-authors describes the protocol of acute cigarette smoke exposure in healthy smokers and smokers with COPD, which, as proposed by the authors, will help monitor local changes in the nasal mucosa and a defined set of systemic responses to this exposure. The major comments are listed below: 1. Although the method is potentially interesting, no data was presented to demonstrate that using this protocol and proposed outcome parameters of cigarette some exposure will provide significant results.2. With regard to the proposed parameters to determine the response to cigarette smoke exposure, the logic behind selection of individual measures and methods is not entirely clear and not always supported by the existing evidence. The protocol could be significantly improved if the authors provide a Table with all evaluation parameters and methods, accompanied by predicted/estimated directions of change, possible interpretation of predicted changes, statistical approaches and references to existing evidence relevant to COPD.3. The authors specify sample size of each group to be evaluated using this protocol. Why and how these numbers were selected? Has this cohort been already recruited and evaluated? Are there any differences between the phenotypes with regard to the selected evaluation parameters at the baseline?4. The "one airway concept" is intriguing, especially with regard to allergic airway diseases, including asthma. However, there is no strong evidence so far supporting relevance of this concept to the biology of COPD, which is currently considered rather a small airway disease. Therefore, preliminary data or literature evidence linking abnormal nasal mucosal biology to COPD would be important.5. The manuscript will benefit from the Table summarizing all advantages and disadvantages, inclusion and exclusion criteria, limitations, potential difficulties and risks for patients.
---

VERSION 1 – AUTHOR RESPONSE

Reviewer Name Renat Shaykhiev

Institution and Country Weill Cornell Medical College

Please state any competing interests or state 'None declared': None declared

The manuscript by Ross and co-authors describes the protocol of acute cigarette smoke exposure in healthy smokers and smokers with COPD, which, as proposed by the authors, will help monitor local changes in the nasal mucosa and a defined set of systemic responses to this exposure. The major comments are listed below:

1. Although the method is potentially interesting, no data was presented to demonstrate that using this protocol and proposed outcome parameters of cigarette smoke exposure will provide significant results.

As this study is extremely novel, it is difficult to assess appropriate number, but below I have provided some evidence that goes some way to justify our choices. We have previously used some of the methods used in this paper in other studies and have seen significant results using low n numbers (3-6). For example we have seen clear delineation in the blood challenge assay between healthy smokers when compared with their age matched controls (Paul-Clark MJ et al., 2008: *Mol Med.* 2008 May-Jun;14(5-6):238-46. doi: 10.2119/2007-00098). This was also true of transcriptomic data from non-smokers PBMCs challenged ex-vivo with cigarette smoke extract (Wright et al., 2012: *PLoS One.* 2012;7(2):e30120. doi: 10.1371/journal.pone.0030120). However, for further reassurance for Dr Shaykhiev, I have attached a principle component analysis plot for the analysis of the transcriptomics data from the blood of all three groups 20 mins post challenge, and as can be seen all smokers and healthy non smokers group together, COPD smokers can be clearly delineated on their overall gene profile at this time point (similar results were obtained for pre and 300min post challenge).

Control Healthy smoker COPD Smoker

2. With regard to the proposed parameters to determine the response to cigarette smoke exposure, the logic behind selection of individual measures and methods is not entirely clear and not always supported by the existing evidence. The protocol could be significantly improved if the authors provide a Table with all evaluation parameters and methods, accompanied by predicted/estimated directions of change, possible interpretation of predicted changes, statistical approaches and references to existing evidence relevant to COPD.

We thank the reviewer for his comments and have included a Table and text defining our criteria in the manuscript.

Summary of Inclusion Criteria and Assessments in the Acute Cigarette Smoke Challenge

Inclusion Criteria

Sample and Assessment Parameters References

General criteria for 3 group of subjects, each of 12 individuals

- 45 to 75 years old
- Good general health, with no chronic illnesses
- No prescribed anti-inflammatory medications (including statins)
- Females of childbearing potential have a negative pregnancy test Plasma:

Cigarette smoke

correlates of exposure

- nicotine
- cotinine
- 3-hydroxycotinine

Plasma mediators

- prostanoids
- metabolites
- cytokines & chemokines

Whole blood ex vivo:

- transcriptomics
- TLR-agonist stimulation of leukocytes
- platelet aggregation Nicotine (9)

Serum cytokines

(10-15)

Serum metabolites

(16-18)

Blood ex-vivo stimulation (19;20)

Smokers:

Current: >5 cigs/day

History: >10 pack yrs

Non-Smokers:

Have never smoked Nasosorption (SAM)

- prostanoids: LTB₄, LTC₄, PGD₂
- metabolites
- cytokines and chemokines

Group 1: Smokers with GOLD Stage 2 COPD

Post-bronchodilator FEV₁ 50-79%; Forced Expiratory Ratio <70%

TLCO of <80% of normal

Emphysema on 5 slice HRCT scan chest Nasal curette

- immunohistology
- transcriptomics
- flow cytometry

epithelial culture

Group 2: Healthy Smokers

Post-bronchodilator FEV₁ ≥80%; Forced Expiratory Ratio ≥ 70%

TLCO ≥ 80% of normal

Normal 5 slice HRCT scan of chest Exhaled breath condensate (EBC) EBC

(21;22)

Group 3: Healthy Non-Smokers

Post-bronchodilator FEV₁ ≥80%; Forced Expiratory Ratio ≥ 70%

TLCO and HRCT not done Sputum Sputum

(23-25)

3. The authors specify sample size of each group to be evaluated using this protocol. Why and how these numbers were selected? Has this cohort been already recruited and evaluated? Are there any differences between the phenotypes with regard to the selected evaluation parameters at the baseline?

We thank the Reviewer for his comment and fully understand his concerns. As already mentioned this study is highly novel in its concept, although we do now have data suggesting that our model is adequately powered, however as the Reviewer and Editor will understand, this data is beyond the remit of this paper.

4. The “one airway concept” is intriguing, especially with regard to allergic airway diseases, including asthma. However, there is no strong evidence so far supporting relevance of this concept to the biology of COPD, which is currently considered rather a small airway disease. Therefore, preliminary data or literature evidence linking abnormal nasal mucosal biology to COPD would be important. We agree with the reviewer and have added referenced text to the manuscript.

Nasal Mucosal Involvement in COPD

The nasal epithelium is the first point in the respiratory system where cigarette smoke has contact with the respiratory mucosa. As part of “the one airway concept” that is well established in asthma, there is also considerable evidence for nasal involvement in COPD (1;2). Patients with COPD have chronic nasal symptoms and impaired quality of life (3-5), with upper and lower airway inflammation (6), and exacerbation of COPD is associated with increased pan-airway inflammation(7). In addition, young “healthy smokers” have functional and inflammatory changes in the nose and lower airways (8).

5. The manuscript will benefit from the Table summarizing all advantages and disadvantages, inclusion and exclusion criteria, limitations, potential difficulties and risks for patients. We agree with the reviewer and have added a table defining advantages and disadvantages of our study.

Features and Ethical issues with the Acute Cigarette Smoke Challenge

Features of the model

Ethical Issues

Advantages:

- Human model
- Acute-on-chronic inflammation
- Serial non-invasive sampling
- Combined direct measurement of biomarkers and ex vivo stimulation
- Limited numbers of strictly defined subjects
- Compare with in vivo animal models

All subjects must be advised to stop smoking, and offered full clinical, psychological and pharmacological support to carry this out.

There must be no encouragement for the subject to begin or continue smoking.

Some frail COPD subjects will have difficulty fasting and refraining from cigarettes for the morning.

Clinical disease detected through the investigations must be fully treated, regardless of participation in the study.

Disadvantages:

- Difficulty recruiting a small number of highly defined subjects
- Need to validate upper versus lower airway inflammation, including tissue biopsies
- Signal parameters must reliably change after acute cigarette smoke exposure
- Lung function and CT changes may occur after acute smoke exposure Some frail COPD subjects will have difficulty fasting and refraining from cigarettes for the morning.

The subject should not be taking any anti-inflammatory or confounding therapy: therapy must not be withheld.